# Transformer-based 3D U-Net with attention blocks for Kidney Tumor Segmentation

Bohdan Petryshak[1,2,3], Danylo Kolinko[3], Igor Krashenyi[4], and Dmytro Fishman[1,2]

[1] Better Medicine OÜ, Tartu, Estonia
[2] Department of Computer Science, University of Tartu, Narva mnt. 18, 51009, Estonia
[3] Ukrainian Catholic University, Ilariona Svjentsits'koho Street, 17, Lviv, 79000, Ukraine
[4] Piñata Farms
dmytro.fishman@bettermedicine.ai

**Abstract.** Here we describe our envisioned solution to The 2021 Kidney and Kidney Tumor Segmentation Challenge which constitutes training the original U-Net architecture with a wide range of modifications, including transformers, residual connections and attention blocks. Once quantitative results are obtained we are happy to modify the present paper.

**Keywords:** segmentation · attention block · transformer

## 1 Introduction

Kidney associated cancer is among the top ten most common cancers in men and women. According to the American Cancer Society the overall risk of kidney cancer is about 1.5%. According to the estimates more than 13,000 people will die as a result of kidney cancer in 2021. World Health Organisation claims that many of these deaths could be prevented through early detection of cancers. Here we present a deep learning system capable of automatically detecting kidney cancers from computed tomography (CT) scans and thus potentially able to improve the outcomes for some of the potential cancer victims. Our present approach is in improving the classical U-Net [6] architecture with several non-trivial modifications such as involution [4], residual connections and attention blocks. Further, we describe our approach in detail as well as give an prognosis of expected results.

## 2 Methods

### 2.1 Training and Validation Data

To increase the amount of training examples, we have used nnU-Net [2] pretrained on KiTS21 data to generate predictions for the KiTS19 dataset [1]. We

have added KiTS19 cases along with generated predictions to already available KiTS19 data. As a result, our final training data consists of 504 unique cases.

We have added the following modifications to the KiTS19 dataset before merging:

1. We have excluded Case IDs 23, 68, 125, 133, 15, and 37 as they have been reported as mislabelled [3].
2. There were only background, kidney, and tumor classes in the KiTS19 competition. We have used nnU-Net baseline model provided by organizers of KiTS21 for relabeling data from KiTS19. We have introduced the cyst class to the KiTS19 dataset and made the task formulation identical to the KiTS21. For the kidney and tumor classes, we have used majority voting between predictions of the nnU-Net model and the ground truth of the KiTS19.

We have validated our approach based on the five-fold cross-validation on merged KiTS19 and KiTS21 datasets. For every fold, we have used random unique 408 cases for training and 102 for validation.

### 2.2   Preprocessing

We have performed a range of pre-processing steps before feeding data into the network.

1. All cases were resampled to the common voxel spacing of $0.781 \times 0.78 \times 0.781$ mm. After resampling, the new median image shape per patient is $528 \times 512 \times 512$ voxels for the training set.
2. We set the level window to the kidney-specific organ HU values range [-79, 304] for every case in the dataset
3. Finally, we do Z-score normalization of the pixel intensity values by subtracting 105 and dividing by 75.

### 2.3   Proposed Method

Based on the de-facto standard approach for dense predictions U-Net, we propose a range of fundamental architectural modifications to improve upon the original idea. We produce three different architectures: residual 3D U-Net, TransBTS [9], attention 3D U-Net and make ensemble out of them.

**Network Architectures** As a basis for all architecture variants, we use 3D U-Net topology. Every model utilizes convolution blocks containing 3D convolution, Leaky ReLU, Dropout, and Instance Normalization. The downsampling operation is done by strided convolution, and upsampling part is achieved via transposed one. For all experiments, we use a network's depth of 5 with a maximum amount of features up to 320. Importantly, the number of convolutional kernels doubles with every downsampling step.

We have not completely settled on a specific configuration of our final approach, yet, we actively testing the following architectures:

*Residual 3D U-Net.* It is a winner architecture [3] of the KiTS19 competition. We assume it should provide the superior performance in this competition as well.

*Transformer-based 3D U-Net.* We try to extend 3D U-Net architecture with a transformer-based component to improve the global (long-range) information modeling using self-attention mechanisms. Currently we test the transformer-based bottleneck (TransBTS) [9], and deformable Transformer (DeTrans) endocer (CoTr) [10].

*Attention-based 3D U-Net.* We are experimenting with the structure of the basic convolutional block by introducing different attention mechanisms to push the neural network accurately concentrate on all the relevant elements of the input improving performance. Currently, we investigate the SE block and its variations [7], PE blocks [5], ECA blocks [8], SA-Net [11], EPSA-Net modules [12].

**Training details** For all our experiments, we use stochastic gradient descent with momentum 0.9. We have trained all our algorithms in a patch-wise fashion. We use the voxel of size $128 \times 128 \times 128$ as an input. The batch size is 2. We train all networks for 1000 epochs with early stopping regularization with the patience of 50 epochs. Cross-entropy and Dice score were used as our main loss function with deep supervision to make the training process smoother. For the training process, we used Nvidia Tesla v100 32GB. It took eight days to train one network configuration and takes 16GB of VRAM on a single GPU. Our implementation is based on nnU-Net [1] framework. We have extensively used all standart augmentation techniques, including random flips, rotations, gamma, brightness, contrast, and random Gaussian noise.

## 3 Results

Unfortunately, no conclusive results have yet been produced on the public datasets as described approach requires massive amounts of time to train on the specialised hardware. We will update this section as soon as we have either results from our cross-validation pipeline or when test set results will be announced.

## Acknowledgment

D.F. was supported by Estonian Research Council grants (PRG1095, PSG59 and ERA-NET TRANSCAN-2 (BioEndoCar)); Project No 2014-2020.4.01.16-0271, ELIXIR and the European Regional Development Fund through EXCITE Center of Excellence.

---

[1] https://github.com/MIC-DKFZ/nnUNet

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
