# OpenReview forum: "Transformer-based 3D U-Net with attention blocks for Kidney Tumor Segmentation"
_MICCAI.org/2021/Challenge/KiTS — Submitted to KiTS21 Challenge_

### Official Review · Reviewer_QV34 · 2021-08-30

**Rating:** 5

**Review:**

The authors present an unfinished paper describing three different approaches to the problem. The paper could really benefit from figures that show visual summaries of each approach so that it is easier to readers to understand what was tried at a glance. The kidney cancer statistics that are cited seem off and should be checked. No "Discussion" section was provided in this initial submission, but the authors should be sure to include one in their revision.

---

### Official Review · Reviewer_vArj · 2021-08-30

**Rating:** 4

**Review:**

### Overall

- The paper makes explicit mention of being unfinished. This is fine for the original submission but please be sure to remove these statements in your revision

### Introduction

- Please cite your sources for the epidemiology numbers and clarify whether they are worldwide or pertain only to a particular country

### Methods

- The KiTS21 dataset uses the same imaging as KiTS19 for cases 0-299. Please rephrase this section where you state the number of "unique" cases
- The KiTS19 labels for cases 23, 68, 125, 133, 15, and 37 have been corrected and no longer need to be excluded
- Given the above, be sure that your training/validation split does not have data leakage
- What strategy did you use for resampling?
- How did you choose the HU range at which to clip values?
- I see you are still deciding between methods. It would be great to keep this information in the final paper and to report the results you got with each of them and explain why you chose your final approach

### Results

- Be sure to add not only the final results but also your cross-validation results for every approach that you tried

### Discussion and Conclusion

- Be sure to add this section in your final paper

---

### Decision · Program_Chairs · 2021-08-30

**Decision:**

Major Revisions

**Comment:**

Please address the reviewer comments and resubmit